# From Waste to Value Added Products: Manufacturing High Electromagnetic Interference Shielding Composite from End-of-Life Vehicle (ELV) Waste

**DOI:** 10.3390/polym16010120

**Published:** 2023-12-29

**Authors:** Roxana Moaref, Shaghayegh Shajari, Uttandaraman Sundararaj

**Affiliations:** 1Department of Chemical and Petroleum Engineering, University of Calgary, Calgary, AB T2L1Y6, Canada; roxana.moaref@ucalgary.ca (R.M.); shaghayegh.shajari@northwestern.edu (S.S.); 2Querrey Simpson Institute for Bioelectronics, Northwestern University, Evanston, IL 60611, USA

**Keywords:** end-of-life vehicle waste, PP/CaCO_3_ (40/60 wt%) master batch, carbon nanotubes, polypropylene, sustainability, electromagnetic interference shielding

## Abstract

The use of plastics in automobiles is increasing dramatically due to their advantages of low weight and cost-effectiveness. Various products can be manufactured by recycling end-of-life vehicle (ELV) plastic waste, enhancing sustainability within this sector. This study presents the development of an electromagnetic interference (EMI) shield that can be used for protecting electronic devices in vehicles by recycling waste bumpers of ethylene propylene diene monomer (EPDM) rubber from ELVs. EPDM waste was added to a unique combination of 40/60: PP/CaCO${}_{3}$ master batch and conductive nanofiller of carbon nanotubes using an internal melt mixing process. This nanocomposite was highly conductive, with an electrical conductivity of $5.2\times 10^{-1}\;S\cdot {\mathrm{cm}}^{-1}$ for 5 vol% CNT in a 30 wt% EPDM/70 wt% PP/CaCO${}_{3}$ master batch and showed a high EMI shielding effectiveness of 30.4 dB. An ultra-low percolation threshold was achieved for the nanocomposite at 0.25 vol% CNT. Waste material in the composite improved the yield strain by about 46% and strain at break by 54% in comparison with the same composition without waste. Low cost and light-weight fabricated composite from ELV waste shows high EMI SE for application in electronic vehicles and opens a new path to convert waste to wealth.

## 1. Introduction

The rapid advancement of fifth generation communication technology in smart vehicles is leading to more electromagnetic interference (EMI), which creates problems such as electronics breakdown, information theft and hardware security vulnerabilities [1]. Moreover, EMI pollution has posed potential threats for living organisms and humans [2,3,4]. Therefore, investigating high-performance EMI shielding materials for smart vehicles has been of great interest for many researchers [5,6,7,8]. To protect sensitive electronic functions, lightweight materials with high electrical and mechanical properties are needed.

Until now, numerous shielding materials have been manufactured that demonstrate effective shielding properties [9,10,11]. Unfortunately, these materials are over-dependent on obtaining a higher reflection of electromagnetic (EM) waves. Reflection of these waves keeps the electromagnetic radiation in the atmosphere and, thus, may affect other devices or reflection into the environment and cause more serious EMI pollution problems [12], often referred to as EMI smog. Conductive polymer composites (CPCs) with high flexibility, corrosion resistance and absorbance-dominant EMI shielding are the next-generation candidates for EMI shielding devices [13]. Another type of polymer well-known for its high EMI shielding effectiveness is an intrinsically conducting polymer (ICP), including polyaniline (PANI), polypyrrole (PPy) and poly(3,4-ethylenedioxythiophene) (PEDOT:PSS) [14,15]. Polymer composites make up a vast portion of the automobile industry due to their light weight and cost-effectiveness. Annually, about 6 million tons of plastic waste are generated from end-of-life vehicles (ELV) [16]. Recycling and reusing plastic resources in ELVs can contribute to a greener environment, increased profit and sustainable development [17,18]. One approach is to fabricate recyclable EMI shielding materials from ELV plastic waste. As shown in Figure 1, polypropylene (PP) and its composites stand out as the most frequently used plastics in automobiles owing to their easy processing, excellent mechanical properties and recyclability [16].

Recycled products suffer from poor mechanical properties due to decreasing molecular weight caused by macromolecular chain scissions when recycling at high temperatures and high shear stresses [16]. To improve their mechanical performance, various fillers can be added to the waste. As an example, inorganic fillers like sodium montmorillonite (clay) [19] and glass fiber [20] can increase the Young’s modulus of the composite. Limestone (calcium carbonate (CaCO${}_{3}$)) is the most abundant and cost-effective filler [21] used in PP to increase the tensile strength, Young’s modulus, dimensional stability, corrosion resistance and design flexibility; however, this filler decreases the ductility of the composite [21,22].

A conductive filler is required for EMI shielding properties; thus, carbon nanotubes (CNTs) with ultra-high strength (10–100 GPa) and high modulus (∼1.0 TPa), and exceptional electrical conductivity ($10^{4}\;$–$10^{6}\;S\cdot {\mathrm{cm}}^{-1}$) and thermal properties (3000–$6000\;W\cdot m^{-1}K$) have been shown to be a highly effective conductive nanofiller for this purpose [23,24,25]. For instance, Verma et al. [24] demonstrated an enhancement in DC conductivity, tensile strength and a high EMI SE value of 47 dB by adding 4.6 vol% CNT to PP random copolymer. Another study [26] showed that using a hybrid filler system (1.75/1.75 vol% CNT/stainless steel fiber) in PP can achieve a high EMI SE of approximately 47.5 dB.

Several studies focused on developing EMI shielding materials by recycling polymer waste. For instance, Xie et al. [27] prepared flexible polymer composites from cross-linked polyethylene (CLPE) cable waste and CNT filler that showed an EMI SE of 35 dB with a tensile strength above 20 MPa and elongation at a failure of 280%. Zhang et al. [28] demonstrated a high EMI SE of 63.2 dB by adding 6.6 wt% of MWCNTs in waste polyurethane foams.

In this work, a recycled high-performance EMI shielding composite was developed using a PP/CaCO${}_{3}$ (40/60 wt%) master batch with CNT filler and ethylene propylene diene monomer (EPDM) bumper waste from ELVs. Firstly, the master batch of PP/CaCO${}_{3}$ (40/60 wt%) was produced by a melt mixing extruder. Then, it was mixed with the nanoscale CNT filler and ethylene propylene diene monomer rubber (EPDM) waste in a batch mixer. The resultant conductive polymer composite (CPC) showed a high EMI SE of about 30.4 dB and improved yield strain of 46% for 5 vol% CNT mixed with a 30 wt% EPDM/70 wt% PP/CaCO${}_{3}$ master batch. The developed methodology resulted in a high-performance, low-cost EMI shielding material, opening up a new path to convert waste to wealth. This approach marks an economical way of manufacturing EMI shields and provides a smart method for reusing waste material in a multi-component polymer composite to achieve an effective shielding with high absorption.

## 2. Experimental

### 2.1. Materials

The master batch of PP/CaCO${}_{3}$ (40/60 wt%) was manufactured by the melt mixing of PP and limestone. A commercially available extrusion grade of PP homopolymer (PP1703), supplied by Pinnacle Polymers (Garyville, LA, USA), was used as the matrix polymer, and limestone (CaCO${}_{3}$ extra fine grind 325) was supplied by Graymont Ltd. (Richmond, BC, Canada). The CaCO${}_{3}$ filler had a minimum purity of 90%, and a fineness of 0.045 mm was used as a filler to increase the recyclability and mechanical properties. PP’s semi-crystalline structure helps the mechanical performance of the overall composite, and PP is widely used in the automotive industry [29,30]. The MWCNTs (Nanocyl™NC7000) purchased from Nanocyl S.A. (Sambreville, Belgium) was used as a conductive filler. Based on the supplier information, the MWCNTs were prepared by the catalytic carbon vapor chemical deposition (CCVD) method. Ground, black-colored waste plastic materials from car bumpers (mainly EPDM) were obtained from PolyCo (Calgary, AB, Canada), and they were washed with reverse osmosis (RO) water. Technical information for PP and all fillers is presented in Table 1.

### 2.2. Composite Preparation

The master batch of PP/CaCO${}_{3}$ (40/60 wt%) was prepared via extrusion by Canada Colors and Chemicals Limited (CCC) Co., Toronto, ON, Canada. For the fabrication of the CPC containing waste and CNT, a Haake Rheomix series 600 batch mixer (Thermo Scientific HAAKE PolyLab OS System, Waltham, MA, USA) fitted with roller blades was used with a broad range of MWCNT loadings (0.1, 0.2, 0.3, 0.6, 1.0, 3 and 5 vol%) and different weight concentrations of waste (0, 5, 15 and 30 wt%). Table 2 illustrates the details of the sample composition. The sample coding is thus: PP describes polypropylene, C denotes CaCO${}_{3}$, W and the number indicate waste and percentage of waste added, respectively, and CNT plus the number following show that CNT is present and what the volume percentage of CNT is. The PP/CaCO${}_{3}$ master batch, waste material pellets and CNT powder were dried in an oven at 60 °C for 12 h. Then, PP/CaCO${}_{3}$ alone was initially fed in the internal mixer for 5 min at 210 °C and 90 rpm. Then, waste material was added and mixed for an additional 5 min. Subsequently, CNT was fed into the internal mixer and mixed for an additional 10 min. The rectangular specimens for EMI shielding and electrical conductivity measurements and circular specimens for rheological measurement, both with 2.0 mm thickness, were molded using a Carver compression molder (Carver Inc., Wabash, IN, USA). The compression molding process was performed at 210 °C for 10 min under a pressure of 40 MPa.

### 2.3. Characterization

Scanning electron microscopy (SEM) was used to obtain the morphological images of the composites. SEM was performed using a high-resolution Philips XL30 equipment (FEI, Hillsboro, OR, USA). Prior to performing SEM, the samples were cryo-fractured in liquid nitrogen. Fourier transform infrared (FTIR) spectroscopy (Agilent Technologies, Santa Clara, CA, USA) was carried out on solid samples with a resolution number of 4 and 32 scans for characterization of different chemical groups in the composite. Differential scanning calorimetry (DSC) was performed to determine the effect of waste and CNT on the overall crystallinity and the nucleation effects. DSC was carried out with a heating–cooling cycle at $10\;K\cdot \min^{-1}$ under nitrogen gas flow at $50\;\mathrm{mL}\cdot \min^{-1}$ using a DSC Q500 (TA Instruments, New Castle, DE, USA). Crystallization and melting temperatures were obtained from DSC data. To investigate thermal properties and mass loss of the samples, thermogravimetric analysis (TGA) using a TA instrument (TGA Q5500) was used. The analysis involved heating the samples at a rate of $10\;{}^{\circ}C\cdot \min^{-1}$ from room temperature up to 700 °C under nitrogen gas flow at $50\;\mathrm{mL}\cdot \min^{-1}$.

Tensile tests were performed using ASTM D638 standard [31] type V samples with a cross-head speed of $1\;\mathrm{mm}\cdot \min^{-1}$ at room temperature. An Instron tensile tester machine (model 5965, Norwood, MA, USA) was used with a 5 KN load cell at room temperature. To estimate the standard deviation in the mechanical tests, at least 5 specimens were tested for each composite.

To measure the electrical conductivity of the samples, two different measurement systems were used. To test samples with electrical conductivity less than $10^{-2}\;S\cdot m^{-1}$, 10 cm diameter circular specimens were molded, and a Keithley (6517A electrometer, Cleveland, OH, USA) connected to a Keithley 8009 test fixture was used. Electrical conductivity higher than $10^{-2}\;S\cdot m^{-1}$ was measured using a Loresta GP resistivity meter (MCP-T610 model, Mitsubishi Chemical Co., Tokyo, Japan) with a four-pin probe using rectangular specimens according to ASTM 257-75 standards [32]. A voltage of 90 V was applied for all the measurements. To measure the standard deviation and the average value, the measurement was repeated for three samples of each composite.

EMI shielding measurements were carried out on rectangular samples ($25\times 15\times 2.5\;{\mathrm{mm}}^{3}$) in the X-band (8.2–12.4 GHz) frequency range using an Agilent E5071C network analyzer (ENA series 300 KHz–20 GHz) attached to a WR-90 rectangular waveguide. A square matrix of scattering parameters was used to calculate EMI shielding effectiveness (EMI SE) and power coefficients:$$\begin{array}{c} \left[\begin{array}{cc} S_{11} & S_{12}\\ S_{21} & S_{22} \end{array}\right]. \end{array}$$ EMI SE is expressed in dB, which is the logarithm of the incident power to the transmitted power ratio [33]. Measurements were repeated for three samples of each composite, and the average value was reported.

The rheological properties of the polymer composite were evaluated with an oscillatory rheometer (MCR 302, Anton Paar, Austria), using parallel plates (PP25) with a 1.8 mm fixed gap at 200 °C. All samples were pre-sheared at 6.28 $\mathrm{rad}\cdot s^{-1}$. To study rheological properties in the linear viscoelastic (LVE) regime, a strain amplitude sweep test was performed at $\gamma_{0}$ = 0.1–1000.0% using an angular frequency of $\omega =6.28\;\mathrm{rad}\cdot s^{-1}$. Based on the dynamic mechanical spectra results, a shear strain of 0.1% was determined to be in the LVE regime, obtaining dynamic mechanical spectra. The storage ($G^{\prime }$), loss ($G^{\prime \prime }$) modulus and complex viscosity ($\eta^{*}$) were recorded as a function of frequency over a 0.1–$600\;\;\mathrm{rad}\cdot s^{-1}$ range at a fixed shear strain value of 0.1%.

## 3. Results and Discussion

### 3.1. Morphology Analysis

Figure 2a,b illustrates the SEM micrographs of PP/C and EPDM waste material. From Figure 2a, we can observe that CaCO${}_{3}$ particles were well dispersed in the PP matrix, even at high loading, and they had a narrow size particle distribution. Figure 2b shows signs of carbon black in the EPDM waste with a very smooth surface. Adding 30 wt% EPDM waste to the PP/C composite with 5 vol% CNT (Figure 2c) led to a good dispersion of both fillers with high interfacial adhesion. EPDM is a noncrystalline polymer with a low glass transition temperature (Tg) of approximately −60 °C, and it is partially miscible in PP [34,35]. Therefore, adding an amorphous polymer to a semi-crystalline polymer (PP) can enhance the processibility of the melt mixing process. The distribution of fillers in this composite was uniform. This can be attributed to two reasons; firstly, EPDM waste rubber decreased the viscosity of the polymer melt, which helped the melt mixing process [34]; secondly, adding CNT applied a high shear stress during the melt mixing process, and fillers could be dispersed better in the composite [36].

Figure 2d–g shows SEM images of PP/C/W15 with different CNT concentrations. They showed the presence of a well-distributed CNT network at all concentrations from 0.6 vol% of CNT in Figure 2d to 5 vol% of CNT in Figure 2g. The formation of a conductive CNT network is necessary for EM wave attenuation in EMI shielding.

### 3.2. Fourier-Transform Infrared Spectroscopy

Figure 3a compares the FTIR spectroscopy of PP/C, EPDM waste, PP/C/W30 composite and PP/C/W30-CNT5 vol%. The band at $2954\;{\mathrm{cm}}^{-1}$ showed a $CH_{3}$ angular deformation, which confirmed the presence of a hydro-carbonic group. Multiple peaks in 2920 and $2850\;{\mathrm{cm}}^{-1}$ were attributed to the asymmetrical stretching vibration and symmetrical stretching vibration of $CH_{2}$, respectively. In EPDM waste, binary sharp peaks at 1463 and $1378\;{\mathrm{cm}}^{-1}$ and one strong peak at $1155\;{\mathrm{cm}}^{-1}$ showed the asymmetrical in-plane bending vibration, symmetrical in-plane bending vibration and wagging vibration modes of $CH_{3}$, respectively [37]. Multiple weak peaks at 2100 ${\mathrm{cm}}^{-1}$ were due to the stretching bond of carbonyl molecules (*C*=*O*), and one sharp peak at $670\;{\mathrm{cm}}^{-1}$ showed the existence of a *C*-*H* group of carbon black in the waste material [38,39]. In PP/C spectra, the carbonate $\left(CO_{3}^{-2}\right)$ peaks at $1410\;{\mathrm{cm}}^{-1}$ (asymmetric *C*-*O* stretching), 873 cm${}^{-1}$ (out-of-plane vibration) and $713\;{\mathrm{cm}}^{-1}$ (in-plane vibration) were attributed to CaCO${}_{3}$ [40]. In PP/C/W30 and PP/C/W30-CNT5 vol% composites, peaks from 1463–1378 cm${}^{-1}$ showed the existence of waste and PP material in the composite. Moreover, the peak at 1410, 873 and $713\;{\mathrm{cm}}^{-1}$ confirmed the presence of CaCO${}_{3}$ in the composites. In PP/C/W30-CNT5 vol%, there were stronger peaks at $2100\;{\mathrm{cm}}^{-1}$ due to the existence of alkene groups in CNT [41].

### 3.3. Thermogravimetric Analysis

In this study, TGA was carried out from room temperature to 900 °C, and mass loss behavior for each composite was observed, which is shown in Figure 3b. A summary of TGA results is shown in Table 3.

A simple (one-step) decomposition profile can be observed for waste from 220 to 480 °C with a mass loss of about 84 wt%. The mass residue of about 16 wt% was attributed to the inorganic materials and ash content of the waste. PP/C, PP/C/W30 and PP/C/W30-CNT5 vol% showed two-step degradation, reflecting two organic structures in the composites, including the PP polymer and CaCO${}_{3}$. For PP/C, the first mass loss occurred from 323 to 480 °C for 40 wt%, corresponding to PP decomposition. The second mass loss between 570–720 °C indicated 22 wt% mass loss and is attributed to CaCO${}_{3}$ [42]. In PP/C/W30 samples, the first mass loss was 49 wt%, starting from 292 to 480 °C, and in the second loss was 15 wt% at 580–700 °C. Therefore, more mass loss can be observed in the PP/C/W30 composite due to the presence of both PP and EPDM. Moreover, the degradation temperature decreased after the addition of waste material, which was because of the lower decomposition temperature of EPDM. In PP/C/W30-CNT5 vol% samples, 42 wt% of mass decomposed from 240 to 450 °C, and in the second decomposition, about 15 wt% happened over the temperature range of 550–720 °C. The reduction of mass loss in this composite was due to the formation of the CNT network, which could increase the heat transfer of the composite through the barrier effect. In this effect, the physical barrier of the CNT network with the matrix inhibits mass loss during heating and decreases the decomposition or evaporation of the components [43].

### 3.4. Differential Scanning Calorimetry (DSC)

The effect of waste and CNT addition on the crystallinity point and heat flow behavior in terms of temperature in PP composites was studied using DSC analysis. DSC thermograms (cooling and heating cycle) of PP/C/W composites before and after the addition of CNT filler and the magnified view of their crystallinity and melting point are shown in Figure 4 and Figure 5, respectively. The crystallization peak ($T_{c}$), the melting temperature ($T_{m}$), the crystallization ($\Delta H_{C}$) and melting enthalpy ($\Delta H_{m}$) are listed in Table 4. Since there are some unknown impurities in the waste material, the calculation of percent crystallinity is not possible; therefore, materials were studied based on their enthalpy and temperature results.

From the results in Figure 4, we postulate that EPDM waste had more nucleation sites than PP/C, and by increasing more waste, both $T_{c}$ and $\Delta H_{C}$ improved. For instance, adding waste from 5 to 30 wt% enhanced $T_{c}$ from 122.3 to 125.0 °C and increased $\Delta H_{C}$ from 36.5 to $45.7\;J\cdot g^{-1}$. Additionally, it was observed that the PP/C composite had a sharper melting point, suggesting the existence of α crystals in the composite, while in waste EPDM, a small melting peak at 126.1 °C suggested the β hexagonal crystalline interfaces that are promoted by the incorporation of nucleating agents [44,45]. Therefore, EPDM waste in the PP matrix could act as a nucleating agent, and therefore, it increased the crystallinity of the composite [46,47,48].

In Figure 5, we see the effect of CNT addition on thermograms, i.e., adding 1 and 5 vol% CNT decreased $\Delta H_{C}$ of the PP/C/W30 composite from 45.7 to 40.0 and $36.0\;J\cdot g^{-1}$, respectively. However, the $T_{m}$ value for the same composites increased from 162.9 to 164.6 and 167.5 °C, respectively. The increase in melting point suggests that CNT can act as a nucleating agent [49] and enhance the formation of the α-crystalline phase in the composite [50]. Additionally, the disappearance of the melting peak at 126.1 °C confirmed this claim.

### 3.5. Rheology

The rheology of the composites was studied within the linear viscoelastic region (LVE). In this region, the structure and the dispersion of the fillers were analyzed without the destruction of the sample. The LVE region was determined through dynamic strain sweep tests, during which the specimens experienced progressively higher oscillatory strain amplitudes while maintaining a constant frequency. This region can be identified as the range of strains at which the $G^{\prime }$ curve versus strain amplitude exhibits a plateau ($\lambda_{L}$) [51].

Figure 6 shows the dynamic modulus ($G^{\prime }$ and $G^{\prime \prime }$) versus strain % for different composites at a fixed frequency of $6.28\;\mathrm{rad}\cdot s^{-1}$. Based on the results, $\lambda_{L}=0.1\%$ is low enough for all composites to be in the LVE region. Based on Figure 6, the addition of CNT changed the predominantly viscous behavior ($G^{\prime \prime }>G^{\prime }$) to a more solid structure ($G^{\prime }>G^{\prime \prime }$) in the LVE region. Apart from this change, both the storage and loss moduli increased after adding CNT. This implies that CNT was likely acting as a reinforcing agent to improve both the strength properties and viscous damping [51]. The cross-over strain (i.e., where $G^{\prime }$ and $G^{\prime \prime }$ are equal) showed the flow point at which the behavior changed from gel to liquid. This change was mainly due to the destruction of the structure at higher strain amplitudes [52]. The cross-over strain increased at a higher CNT vol%; for instance, in Figure 6a, the cross-over strain shifted from 7.1% for PP/C-CNT1 vol% to 31.9% for PP/C-CNT5 vol%. This represented the formation of the CNT network at a higher CNT concentration. On the other hand, adding waste material decreased the cross-over point, as shown in Figure 6a,d. This means that the composites with higher waste content tended to flow and shift to a liquid-like behavior at lower strains due to the viscous characteristics of the EPDM waste.

Figure 7 illustrates the oscillatory frequency sweep test at $\lambda_{L}=0.1\%$ on the composites and can be used to analyze the time-dependent behavior of the samples in the non-destructive deformation range. Regarding Figure 7a,c,e,g, for samples containing CNT, the storage modulus was always higher than the loss modulus in all frequency ranges because of the CNT network. Additionally, the solid-like behavior ($G^{\prime }$) improved for a higher CNT content. Composites without CNT showed a change at low frequencies from an elastic to viscous behavior because the composite structure was destroyed at higher frequencies. Figure 7b,d,f,h illustrates the complex viscosity of the composites. The composites showed a shear thinning effect, whereas the samples without CNT were less affected by the frequency. For a higher CNT vol%, the viscosity was higher for all composites, showing the reinforcing role of the CNT filler. Comparing Figure 7b,d, the complex viscosity increased for PP/C-CNT5 vol% by adding 5 wt% waste material, while for other composites, the viscosity decreased at a higher waste percentage. This finding might be due to the improvement in the CNT network structure at this vol% by adding 5 wt% waste material.

### 3.6. Mechanical Properties

The effect of waste addition on the stress–strain curve for composites with and without CNT is shown in Figure 8a,b, respectively. Based on Figure 8a, waste material had the highest ductility over composites. Adding waste up to 30 wt% enhanced the toughness of the composite and strain to failure of PP/C from 1.1 to 1.7%. However, waste material showed the lowest modulus due to its rubber-like behavior, and composites with a higher amount of waste demonstrated a lower modulus. This behavior was seen in the rheology results when the dynamic modulus versus strain decreased after adding waste. Figure 8b depicts the mechanical properties of the composites after adding 5 vol% CNT. Adding CNT increased the tensile strength (TS) at failure and toughness of the composites. For example, for PP/C/W5, TS at failure increased from 14.45 MPa to 21.32 MPa upon the addition of 5 vol% CNT.

Figure 9 shows the tensile test data of different composites, including the modulus, tensile strength at failure, yield stress and yield strain. Figure 9a shows a decrease in the modulus of PP/C with waste addition from 5 to 30 wt% but an increase in modulus by adding CNT filler to PP/C and PP/C/W composites. For instance, by adding 5 wt% of waste material, the modulus of PP/C decreased by 13.6%, while 5 vol% CNT improved the modulus of PP/C/W5 by 28%. Improvement of the elastic modulus in rheology and tensile tests confirms that adding CNT boosts the mechanical performance of recyclable waste materials. Moreover, modulus enhancement could be attributed to the higher crystallinity after CNT addition [53], according to DSC results. Figure 9b shows that by adding a small amount of waste material, the TS of PP/C composite increased significantly because EPDM waste remained as small, dispersed particles that contributed to the increase in the interfacial adhesion of fillers to the matrix [54]. Another increase occurred when adding CNT to PP/C/W composites, with the highest value for PP/C/W5-CNT5 (21 MPa), which indicated a further reinforcement effect of CNT. Similarly, Figure 9c illustrates that yield stress improved in the first step with the addition of waste to PP/C, and the second step increase occurred with the addition of 5 vol% CNT to the PP/C composite. The improvement in yielding properties is evidenced by a higher cross-over frequency in the rheology results of the samples with more CNT content. In Figure 9d, strain at the break of PP/C decreased for the PP/C/W5 composite but then improved with the further addition of EPDM waste. This might be due to the crazing effect for toughening PP/C; smaller EPDM particles were ineffective for craze formation. As EPDM content increased, shear yielding and crazing were enhanced, and this imparted greater break resistance [55]. Adding 5 vol% CNT, together with waste addition, enhanced the strain at the break significantly due to the ductile nature of EPDM waste and the reinforcement effect of CNT. From the results in Figure 9, it can be concluded that the mixture of CNT and EPDM waste could improve mechanical properties; moreover, this enhancement showed a good dispersion of CNT in the composite.

### 3.7. Electrical Conductivity

Electrical conductivity is important for many polymer composite applications [6]. The formation of the CNT network in different composites with different waste content (0, 5, 15 and 30 wt%) was measured using DC electrical conductivity and percolation curves, as shown in Figure 10a. The percolation threshold refers to the critical point at which an insulate material becomes a conductor by adding conductive fillers. Based on percolation theory, the electrical conductivity (σ) of polymer composites is related to the volume fraction of the conductive fillers (ϕ) based on the following equation [56,57]:(1)$$\sigma \propto {\left(\phi -\phi_{c}\right)}^{t},$$
where $\phi_{c}$ is the the percolation threshold, and *t* is the power-law constant, which is related to the intrinsic conductivity of the filler. This equation can be fitted to the conductivity data in terms of filler volume percentage. By fitting Equation (Equation 1) (Figure 10a), the percolation thresholds were calculated for PP/C, PP/C/W5, PP/C/W15 and PP/C/W30 for 0.21, 0.22, 0.23 and 0.25%, respectively. By adding waste to PP/C, the electrical conductivity reduced and percolation threshold increased due to the insulation effect of waste EPDM. The electrical conductivity as a function of waste concentration is illustrated in Figure 10b for different CNT vol%. Generally, there was a decreasing trend in the conductivity with increasing waste concentration. At 3 vol% CNT, having 5 wt% waste increased the conductivity from $2.8\times 10^{-1}$ to $4.5\times 10^{-1}\;S\cdot {\mathrm{cm}}^{-1}$, followed by a decrease at higher waste concentrations. The reason for improved conductivity in the PP/C/W5 composite at 3 vol% CNT lies in the stronger CNT network that formed at lower viscosity. As shown in Figure 7, the complex viscosity in PP/C-CNT 3 vol% decreased after having 5 wt% of waste. This decrease in viscosity facilitated the movement and arrangement of CNT filler to form a conductive path for electron transport. Waste content that was 5 wt% reduced the conductivity due to the higher amount of insulating material [58,59].

In a study performed by Stan et al. [60], recycled high density polyethylene (HDPE) with 5 wt% MWCNT showed $0.135\;S\cdot {\mathrm{cm}}^{-1}$ conductivity; also, in another study by Sandu et al. [61], the conductivity of the recycled ethylene-vinyl acetate with 5 wt% CNT was reported to be $0.1\;S\cdot {\mathrm{cm}}^{-1}$. In our study, the conductivity of the composites with 5 wt% waste and 5 vol% CNT was one order of magnitude higher than previously reported data, which was due to better distribution of the conductive filler in the polymer matrix.

### 3.8. Electromagnetic Interference Shielding

#### 3.8.1. Theory of EMI Shielding

Polymer composites containing mobile charge carriers, electric or magnetic dipoles have been extensively studied for their potential use in electromagnetic interference (EMI) shielding [6,62,63]. In a material containing conductive filler, shielding happens due to a Faraday cage cavity as free electrons distribute on the shell of the material, and they cancel the field’s effect inside the cage (Figure 11a) [6,63]. This distribution increases the reflection of EM waves due to higher impedance mismatch between the external space and the shielding material. The remaining EM waves reflect many times between layers inside the shield (multiple reflection) until they are fully absorbed; this dissipation is in the form of heat, and it is called absorption [64,65,66]. Absorption happens due to ohmic loss, dielectric loss, magnetic loss and destructive interference by the incident and reflected EM waves [6,67]. Figure 11b shows the three main mechanisms of shielding, including reflection, absorption and multiple-reflection [68]. EMI parameters according to the shielding mechanism are described as the following equation:(2)$${\mathrm{SE}}_{T}={\mathrm{SE}}_{R}+{\mathrm{SE}}_{A}+{\mathrm{SE}}_{M},$$
where SE${}_{T}$, SE${}_{R}$, SE${}_{A}$ and SE${}_{M}$ show the EMI SE of the total, absorption, reflection and multiple reflection, respectively. EMI SE values are calculated from the transmittance (T) and reflectance (R) of EM waves, which are measured directly by the network analyzer, and absorbance (A) can be calculated given that constant incident power (I=1mW) was used. These power coefficients represent the contribution of absorption and reflection to the overall shielding [25,69]. The following equations quantify EMI SE parameters based on coefficient values:(3)A+R+T=1
(4)$${\mathrm{SE}}_{A}=-10\log\left(\frac{T}{1-R}\right)$$
(5)$${\mathrm{SE}}_{R}=-10\log\left(1-R\right).$$ Herein, *A*, *R* and *T* are the EMI shielding power coefficients for absorption, reflection and transmittance, respectively [69].

The properties affecting the absorption of EM waves are thickness (*t*), electrical conductivity (σ), complex permeability (μ) and permittivity (ε) [70]. In electromagnetism, the magnetization of a material in response to a magnetic field is called permeability [71], and the ability of a dielectric for electric polarization is called permittivity [72]. Permittivity ($\varepsilon ,\;F\cdot m^{-1}$) is the ratio of electric flux density ($D,\;C\cdot m^{-2}$) to electric field intensity ($E,\;V\cdot m^{-1}$). ε is a complex function with real ($\varepsilon^{\prime }$) and imaginary ($\varepsilon^{\prime \prime }$) parts. The real part shows the displacement current that is related to the polarization effect, and it represents the storage capability of the material. The imaginary part illustrates conduction currents due to mobile charge carriers, and it characterizes the dissipation ability of the material. The relationships are given by the following equations [73,74]:(6)$$D=\varepsilon \left(\omega \right)E$$
(7)$$\varepsilon \left(\omega \right)=\varepsilon^{\prime }\left(\omega \right)-i\varepsilon^{\prime \prime }\left(\omega \right)=\left|\frac{D_{0}}{E_{0}}\right|\left(\cos\delta -i\sin\delta \right),$$
where δ is the loss angle, and $D_{0}$ and $E_{0}$ are the amplitudes of the displacement and electric fields, respectively.

#### 3.8.2. EMI Shielding Results of the Composites

Figure 12 shows the EMI SE and power coefficients as a function of the CNT concentration. Based on Figure 12a,c,e,g, the EMI SE for all composites showed an increasing trend after CNT addition because of more mobile charges in the composite, which resulted in a higher reflection of EM waves [75]. Higher waste content decreased the EMI SE of the composites, for example, in 5 vol% CNT, SE${}_{T}$ decreased from 41 in PP/C to 30 dB in PP/C/W30. As shown in Figure 12a, for PP/C samples, both SE${}_{R}$ and SE${}_{A}$ increased after adding CNT, while for samples containing waste material, a slight reduction in the SE${}_{R}$ value occurred at 1 vol% CNT (Figure 12c,e,g).

To explore the contribution of absorption and reflection to the overall shielding, power coefficients in the X-band frequency range were studied. Based on Figure 12b,d,f,h, generally, reflectance increased at a higher CNT concentration, which correlated with higher conductivity. In samples containing waste (Figure 12d,f,h), a decrease in reflectance at 1 vol% was seen. This behavior in reflectance was reported previously by Al-Saleh and Sundararaj [25]. Based on Equation (Equation 5), the reason for a lower SE${}_{R}$ value at 1 vol% CNT was decreased reflectance in the same composites. These composites possessed the highest absorbance among others.

EMI shielding effectiveness is a relative measure, and it is unrelated to absolute power values [25,76,77,78]. Based on coefficient values, at CNT concentrations lower than 3 vol% CNT, absorbance is the dominant mechanism, while at a higher CNT vol%, the reflectance mechanism is dominant. At a higher CNT vol%, EMI SE increased significantly due to higher reflectance. For instance, the EMI SE of PP/C/W30 improved from 11.6 dB at 1 vol% CNT to 30.4 dB for 5 vol% CNT, which was in accordance with the increase in reflectance from 0.24 to 0.65. Therefore, we can conclude that a higher CNT vol% resulted in reflectance-dominant shielding.

The X-band frequency variation of the real and imaginary permittivity for the composites are shown in Figure 13. In the composites without CNT filler, neither $\varepsilon^{\prime }$ or $\varepsilon^{\prime \prime }$ changed for different waste amounts and were relatively independent of frequency. Both $\varepsilon^{\prime }$ and $\varepsilon^{\prime \prime }$ increased with the addition of CNT, and both showed relatively high values for real and imaginary permittivity compared to pure polymer materials. This shows the ability of CNT to significantly change dielectric properties of the composites. Based on real permittivity plots, at a higher CNT concentration, real permittivity increased remarkably, with the exception of $\varepsilon^{\prime }$ in Figure 13e. In general, $\varepsilon^{\prime }$ was independent of frequency. For all composites, $\varepsilon^{\prime \prime }$ increased notably at 5 vol% CNT, showing high values with a decreasing trend with increasing frequency. This shows that at a higher CNT concentration, there is greater dissipation/loss due to a higher number of dielectric charges. By comparing imaginary permittivity plots for different waste material, $\varepsilon^{\prime \prime }$ decreased by adding waste, which shows that the waste material has insulating properties.

## 4. Conclusions

In this research, for the purpose of EMI shielding, composites containing ELV waste and a master batch with 60 wt% of CaCO${}_{3}$ and different vol% of CNT filler were fabricated. The introduction of ELV plastic waste and a high content of CaCO${}_{3}$ increased the sustainability of the system. It was observed that PP/C/W30 with 5 vol% CNT prepared by a simple melt mixing process exhibited excellent conductivity and EMI shielding properties, as well as good tensile behavior. Based on findings from SEM and DSC, PP/C/W30-CNT5 vol% had good properties due to a better dispersion of fillers and high crystallinity in the polymer composite. In fact, a high modulus CNT in this composite improved the degree of crystallization and provided higher tensile strength. In addition, ductile EPDM waste increased strain at the break and yielding of the composite. In terms of the EMI shielding mechanism, the interconnected network of CNT in the composite induced a high reflection and acceptable absorption of EM waves. PP/C/W30 with 1 vol% CNT had a high absorbance of EM power but had low SE${}_{T}$ value (lower than 20 dB), while the PP/C/W30-CNT5 vol% composite with the highest levels of waste material exhibited a high SE${}_{T}$ of 30.44 dB and 65% reflectance of total EM power. Thus, it was the most favourable. The recycled conductive composites in EMI shielding materials with improved mechanical properties can be used for the protection of electronics devices in vehicles. This study presented an innovative approach for repurposing ELV waste in the EMI shielding application, offering an environmentally friendly and economically viable solution for next-generation electronic devices in both academic and industrial contexts.

## Figures and Tables

**Figure 1 polymers-16-00120-f001:**
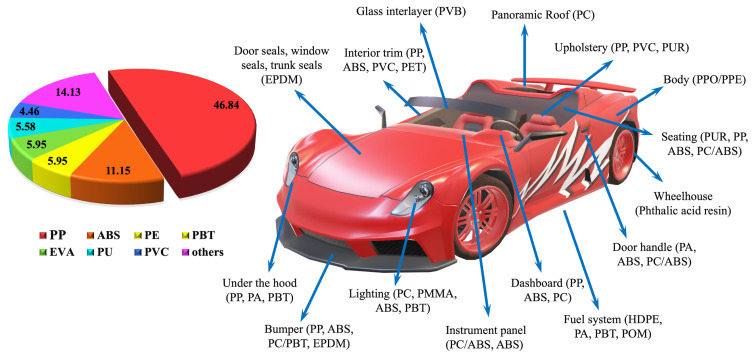
Different plastic parts used in automobiles with the usage percentage [16].

**Figure 2 polymers-16-00120-f002:**
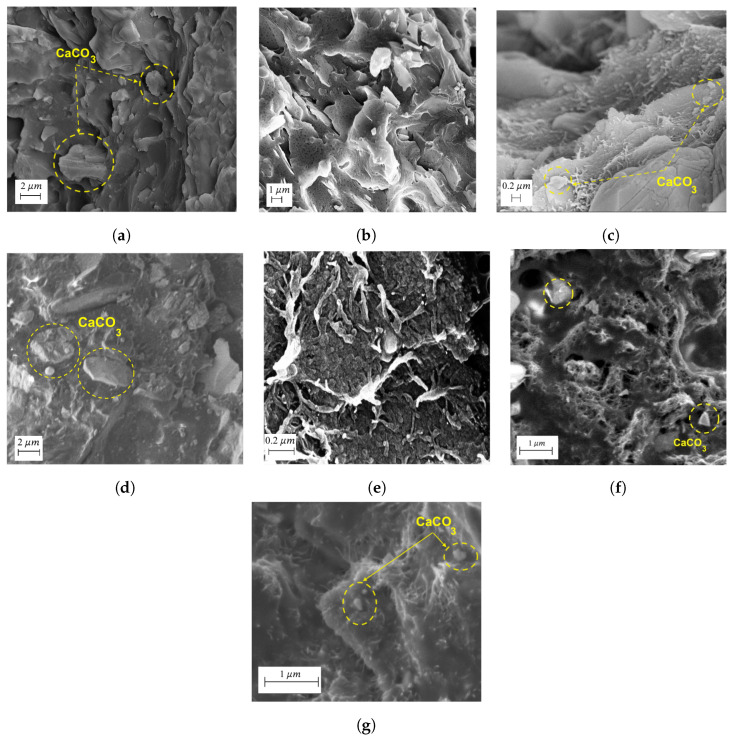
Surface morphology study results: SEM images of PP/C composite (**a**), waste material (**b**), PP/C/W30-CNT5 (**c**), PP/C/W15 composites with different CNT content at 0.6 vol% (**d**), 1 vol% (**e**), 3 vol% (**f**) and 5 vol% (**g**).

**Figure 3 polymers-16-00120-f003:**
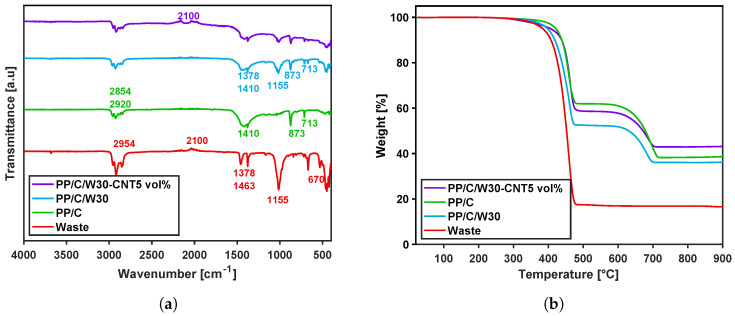
FTIR spectra: FTIR spectra of PP/C, waste, PP/C/W30 and PP/C/W30-CNT5 vol% (**a**). TGA results: TGA plot of PP/C, waste, PP/C/W30 and PP/C/W30-CNT5 vol% at a rate of $10\;[{}^{\circ}C\cdot \min^{-1}]$ (**b**).

**Figure 4 polymers-16-00120-f004:**
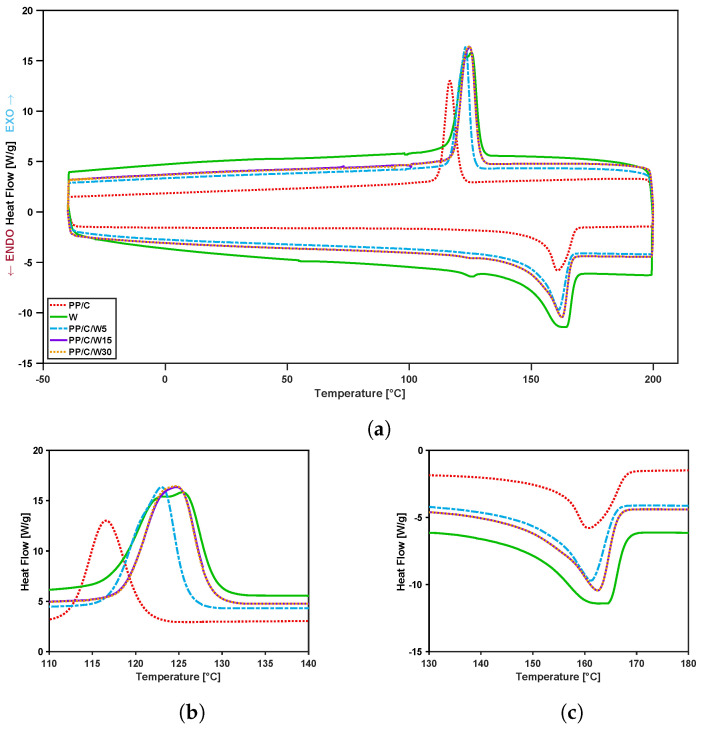
DSC curves of composites: DSC curve at different weight percentage of waste (**a**), crystallinity point (**b**), melting point (**c**).

**Figure 5 polymers-16-00120-f005:**
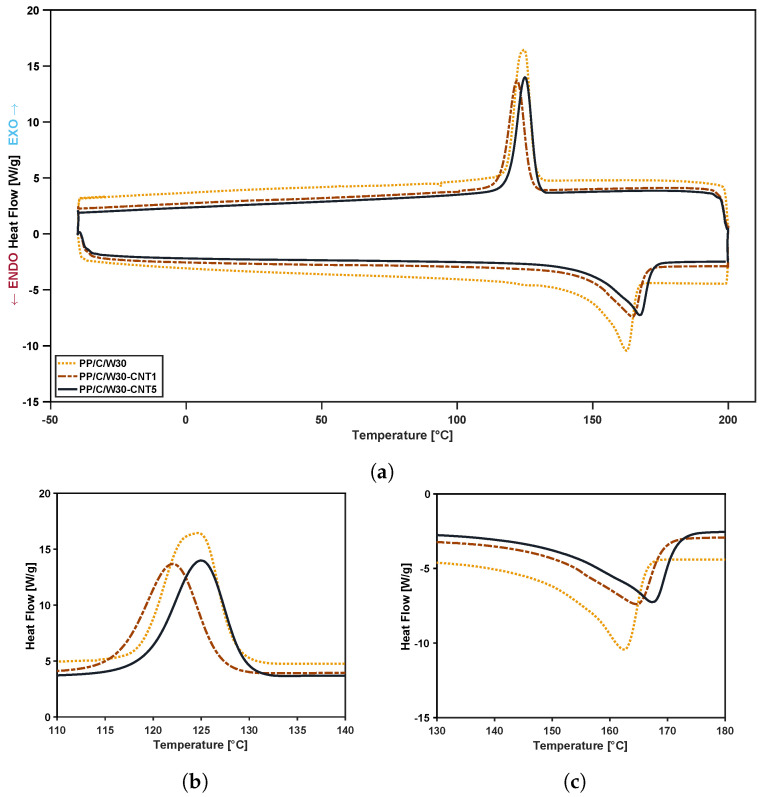
DSC curves of composites: DSC curve of PP/C/W30 composite and its composites at 1 and 5 vol% CNT (**a**), crystallinity point (**b**), melting point (**c**).

**Figure 6 polymers-16-00120-f006:**
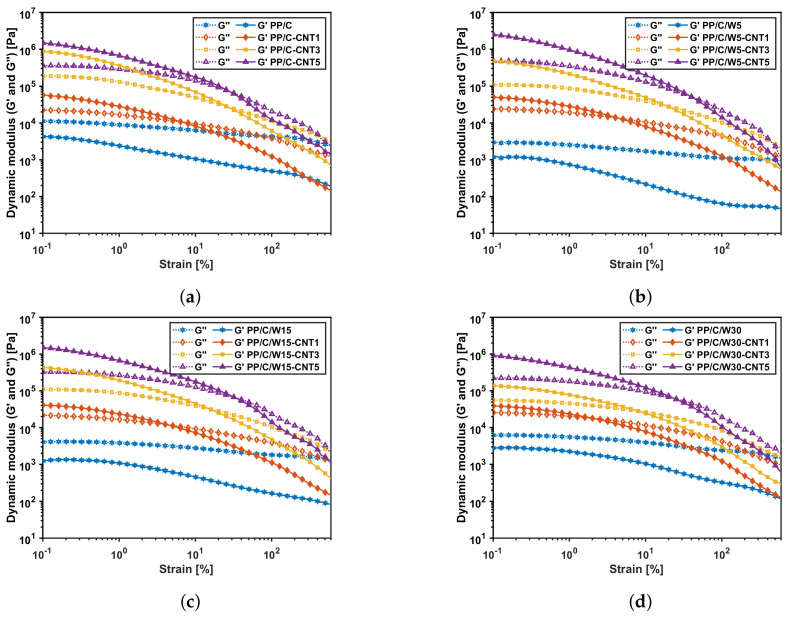
Strain sweep rheology results: storage and loss modulus versus shear strain of the PP/C composite (**a**), PP/C/W5 composite (**b**), PP/C/W15 composite (**c**) and PP/C/W30 composite (**d**) at a fixed frequency of 6.28 $\mathrm{rad}\cdot s^{-1}$.

**Figure 7 polymers-16-00120-f007:**
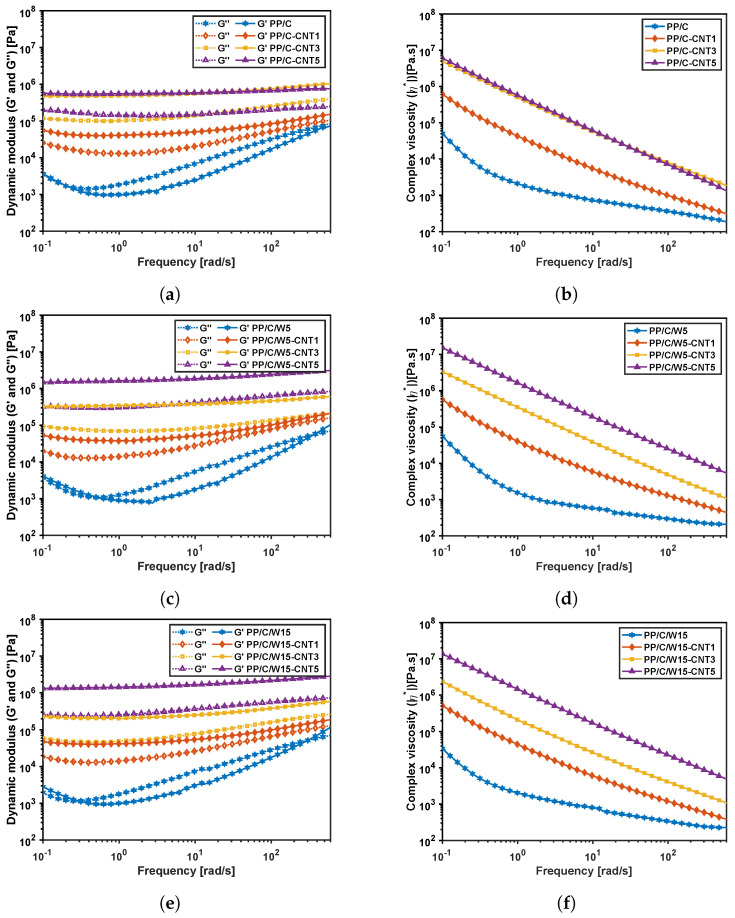
Frequency-sweep rheology results: dynamic modulus and complex viscosity versus the frequency of the PP/C composite (**a**,**b**), PP/C/W5 composite (**c**,**d**), PP/C/W15 composite (**e**,**f**) and PP/C/W30 composite (**g**,**h**) at a fixed shear strain of 0.1%.

**Figure 8 polymers-16-00120-f008:**
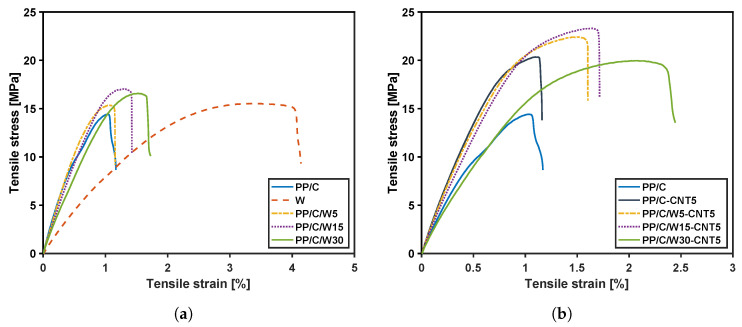
Mechanical properties: stress–strain plot of the composites with 1 mm/min speed and at room temperature without (**a**) and with CNT (**b**).

**Figure 9 polymers-16-00120-f009:**
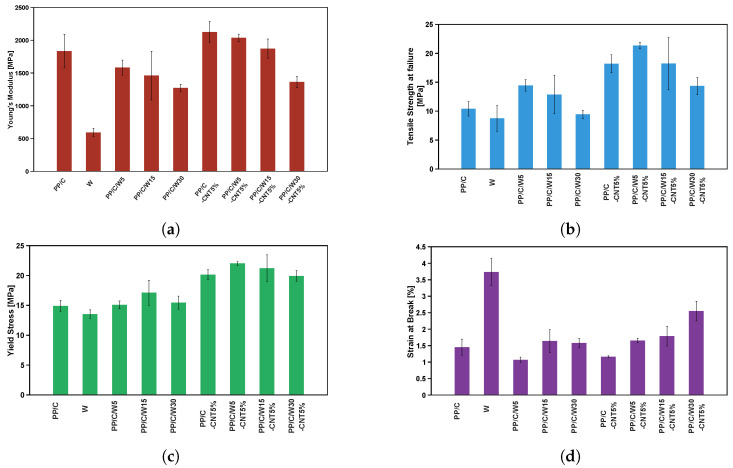
Mechanical properties of different composites: Young’s modulus (**a**), tensile strength at failure (**b**), yield stress (**c**) and strain at the break (**d**) as a function of waste and CNT loading.

**Figure 10 polymers-16-00120-f010:**
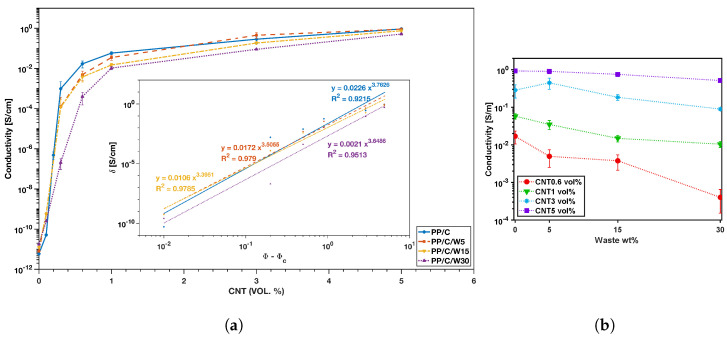
Conductivity results: percolation threshold plots; the inset shows the threshold based on the equation (**a**), conductivity plot versus waste wt% in various CNT vol% (**b**).

**Figure 11 polymers-16-00120-f011:**
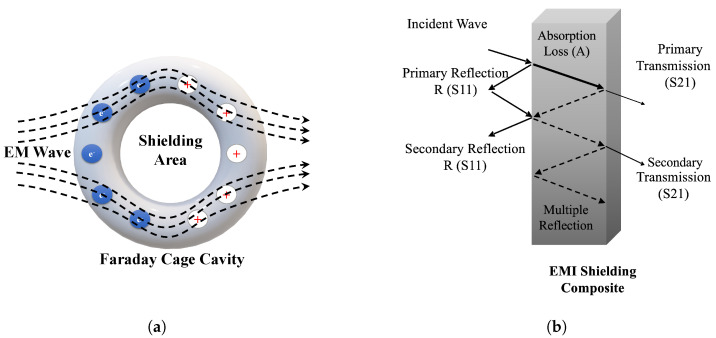
Faraday cage cavity in a conductive material (**a**); transmission line theory describing different mechanisms of shielding (**b**).

**Figure 12 polymers-16-00120-f012:**
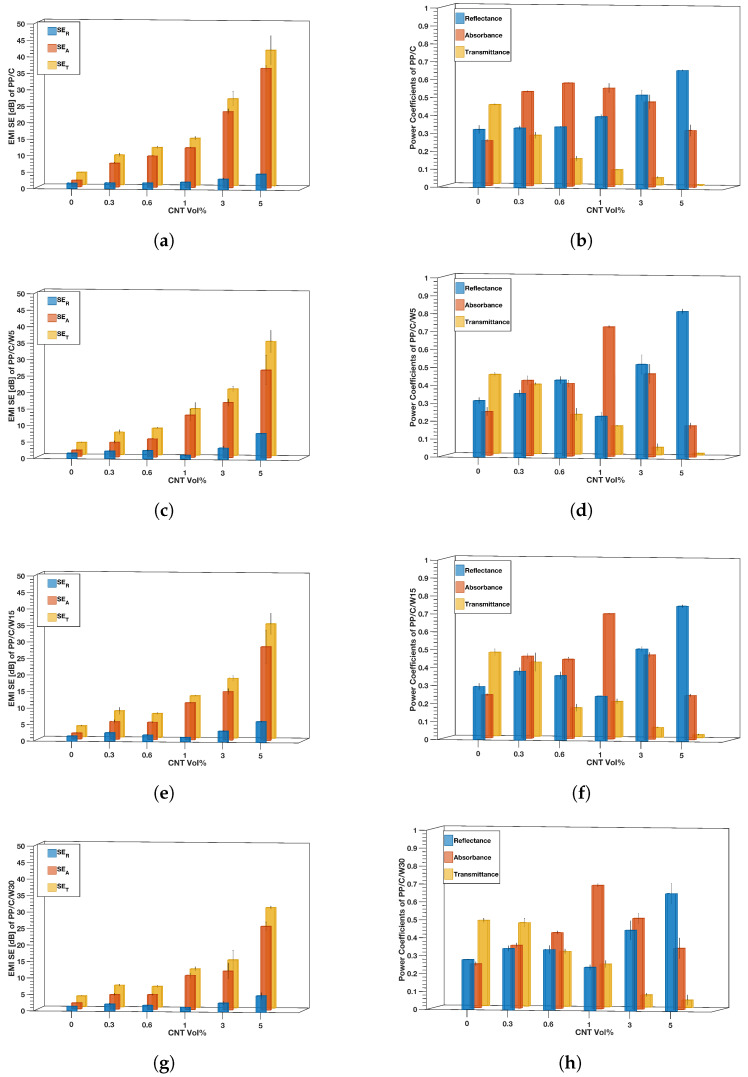
EMI shielding results: EMI SE and EMI power coefficients in terms of CNT addition for different composites, PP/C composite (**a**,**b**), PP/C/W5 composite (**c**,**d**), PP/C/W15 composite (**e**,**f**) and PP/C/W30 composite (**g**,**h**).

**Figure 13 polymers-16-00120-f013:**
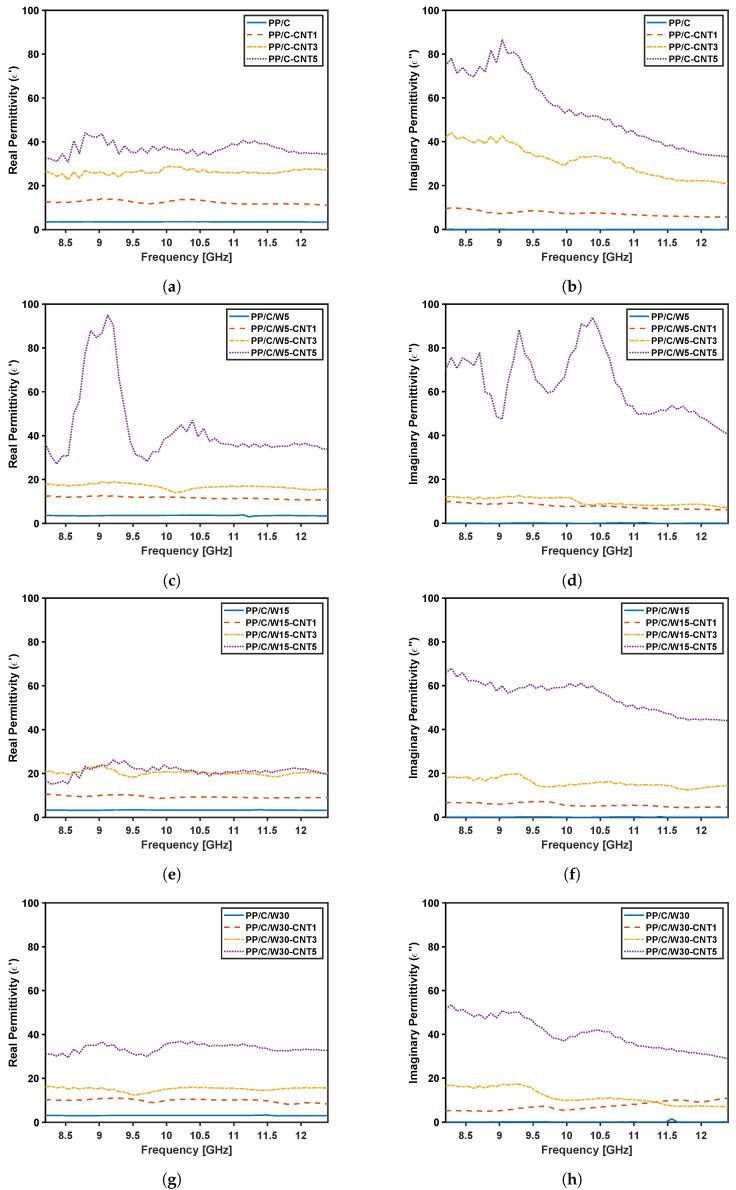
Permittivity results: real and imaginary permittivity values for different composites versus frequency for a PP/C (**a**,**b**), PP/C/W5 (**c**,**d**), PP/C/W15 (**e**,**f**) and PP/C/W30 composite (**g**,**h**).

**Table 1 polymers-16-00120-t001:** Technical information of materials used in the experiments.

Materials	Density [g· cm${}^{-3}$]	Melt Flow Index [g/10 min, 230 °C/2.16 kg]	Average Diameter [μm]	Average Length [μm]	Specific Surface Area [m${}^{2}\cdot g^{-1}$]
Polypropylene	0.905	3.5	N/A	N/A	N/A
MWCNT	1.75	N/A	9.5	1.5	250–300
Limestone (CaCO${}_{3}$)	2.71	N/A	0–45	N/A	0.429
ELV waste	1.12	N/A	N/A	N/A	N/A

**Table 2 polymers-16-00120-t002:** Composition of different samples.

Sample Name	Polypropylene (wt%)	CaCO${}_{3}$ (wt%)	Waste (wt%)
PP/C	40	60	0
PP/C/W5	38	57	5
PP/C/W15	34	51	15
PP/C/W30	28	42	30
W	0	0	100

**Table 3 polymers-16-00120-t003:** Mass loss of the composites derived from TGA analysis.

Sample Name	First Mass Loss [%]	Temperature Range [°C]	Second Mass Loss [%]	Temperature Range [°C]	Remaining Mass [%]
PP/C	40	323–480	22	570–720	38
W	84	220–480	N/A	N/A	16
PP/C/W30	49	292–480	15	580–700	36
PP/C/W30-CNT5	42	240–450	15	550–720	43

**Table 4 polymers-16-00120-t004:** Thermal properties of PP/C composites from DSC analysis.

Sample Name	$T_{m}\;\left[{}^{\circ}C\right]$	$T_{c}\;\left[{}^{\circ}C\right]$	$\Delta H_{m}\;\left[J\cdot g^{-1}\right]$	$\Delta H_{c}\;\left[J\cdot g^{-1}\right]$
PP/C	162.4	115.1	29.0	30.9
W	164.8	126.0	40.9	51.9
PP/C/W5	162.9	122.3	32.0	36.5
PP/C/W15	163.3	122.3	33.1	41.1
PP/C/W30	162.9	125.0	38.1	45.7
PP/C/W30-CNT1	164.6	122.1	34.8	40.0
PP/C/W30-CNT5	167.5	124.9	37.1	36.0

## Data Availability

The data presented in this study are available on request from the corresponding author.

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
