# Peer review of "From Waste to Value Added Products: Manufacturing High Electromagnetic Interference Shielding Composite from End-of-Life Vehicle (ELV) Waste"

_polymers, 2023, doi:10.3390/polym16010120_

Round 1
Reviewer 1 Report
Comments and Suggestions for Authors
Dear Authors,
Thank you for your interesting manuscript. I believe it can be suitable for publication after revision.
1. The unnecessary abbreviations are better to be excluded for the abstract or have to be introduced when firstly used (e.g. FTIR, SEM...). Also the abbreviations have to be excluded from the keywords (e.g. ELV, CNT, PP).
2. The text contains some notations which are need to be reformatted according to the MDPI requirments, e.g. "S.cm", "10˘100 GPa", "8.2˘12.4 GHz", etc.
3. I suppose it is better to add some information on the intrinsic conductive polymers (e.g. polyaniline, PEDOT:PSS, etc.), since these class of materials are actively used for EMI shielding and absorbing.
4. Line 55, "can increase modulus of the composite", "Young's" seems to be omitted.
The same is for Figure 9. Now it is only "Modulus [MPa]".
5. All the suppliers and manufacturers information (company name, country of origin) needs to be added in the Experimental section.
6. Line 94, the method and equipment for grinding need to be specified.
7. In the "2.3 Characterization section", the form and the size of the sample for EMI shielding measurements need to be specified.
8. Table 3, the degrees of Celsius symbols need to be corrected.
9. Table 4, the same for the the crystallization and melting enthalpy. Now they are only the black squares instead of the Deltas.
10. The text in the inset in Figure 10 is too small to be read.
11. All the figure captions need some generalizing part, e.g. "Figure 0. Surface morphology studies results: (a) optical microscopy images; (b) electron microscopy images".
12. Figure 11, the right panel, the second "Primary Transmission (S21)" needs to be changed to "Secondary Transmission (S21)".
13. Figure 13, "Permittivity" needs to be changed to "Real Permittivity".
14. The complex permittivity parts values, the real and the imaginary, are quite strange: the real permittivity for some samples is less than 1, the imaginary permittivity for some samples is less than 0. In my opinion, the calculated results for the complex permittivity are not correct. If the Authors are sure it is not so, they are recommended to give some explanations for so strange results basing on the literature data. Nevertheless, I am quite convinced the permittivity calculations are not correct and need to be repeated.
Reviewer 2 Report
Comments and Suggestions for Authors
The authors produced recycled composites for electromagnetic shielding, using carbon nanotubes to conduct electrical current. The manuscript is important because it reuses waste, adding value and generating application potential. In addition, the manuscript is well written and grounded in the literature. Some small revisions:
> Abstract. Line 23-24. “These materials were characterized with FTIR, SEM, DSC, TGA, and an oscillatory rheometer.” Please delete this sentence;
> Page 2. Line 71-72. Please indicate the contribution of the manuscript to the academic and industrial sector, justifying the publication;
>Composite Preparation. State the type of rotor used in the internal mixer;
> Characterization. FTIR inform the resolution and scan amount. What type of sample (powder, film, etc.)?
Comments on the Quality of English Language
Minor editing of English language required
Round 2
Reviewer 1 Report
Comments and Suggestions for Authors
Dear Authors,
Thank you for addressing my comments.